# Size Effect on the Post-Necking Behaviour of Dual-Phase 800 Steel: Modelling and Experiment

**DOI:** 10.3390/ma16041458

**Published:** 2023-02-09

**Authors:** Lintao Zhang, Will Harrison, Shahin Mehraban, Stephen G. R. Brown, Nicholas P. Lavery

**Affiliations:** Future Manufacturing Research Institute, College of Engineering, Swansea University, Bay Campus, Fabian Way, Swansea SA1 8EN, UK

**Keywords:** dual-phase steel, aspect ratio, necking modes, fracture angle, rapid alloy prototyping

## Abstract

This work investigated the feasibility of using a miniaturised non-standard tensile specimen to predict the post-necking behaviour of the materials manufactured via a rapid alloy prototyping (RAP) approach. The experimental work focused on the determination of the Lankford coefficients (r-value) of dual-phase 800 (DP800) steel and the digital image correlation (DIC) for some cases, which were used to help calibrate the damage model parameters of DP800 steel. The three-dimensional numerical simulations focused on the influence of the size effect (aspect ratio, AR) on the post-necking behaviour, such as the strain/stress/triaxiality evolutions, fracture angles, and necking mode transitions. The modelling showed that although a good correlation can be found between the predicted and experimentally observed ultimate tensile strength (UTS) and total elongation. The standard tensile specimen with a gauge length of 80 mm exhibited a fracture angle of ∼55°, whereas the smaller miniaturised non-standard specimens with low *ARs* exhibited fractures perpendicular to the loading direction. This shows that care must be taken when comparing the post-necking behaviour of small-scale tensile tests, such as those completed as a part of a RAP approach, to the post-necking behaviours of standard full-size test specimens. However, the modelling work showed that this behaviour is well represented, demonstrating a transition between the fracture angles of the samples between 2.5 and 5. This provides more confidence in understanding the post-necking behaviour of small-scale tensile tests.

## 1. Introduction

Dual-phase (DP) steels are used widely in the automotive industry, and 74% of the steels used in the field are DP grade [1]. The application of DP steels can increase passenger safety due to their high tensile strengths, even in thin sections, and also can contribute positively to environmental factors, such as weight, fuel consumption, and carbon footprint, thanks to their lightness. The first patent for dual-phase steel was filed in the USA in 1968 [2] and achieved further development in the 1970s: it formed a new category of high-strength low-alloy (HSLA) steels with a higher level of tensile strength that presents an ‘unusual’ good ductility [3]. Hayami and Furukawa first used the expression ‘dual-phase steel’ to emphasise the heterogeneous microstructure, compared to the mild steels thought to be single-phase [4]. The term ‘dual phase’ refers to the ferrite and martensite phases, and this combination presents better mechanical properties: a higher tensile strength (enabled by the martensite) and a higher ductility, compared to ferritic–pearlitic steel. The main chemical elements of DP steel are C (0.06–0.15 wt%, strengthening the martensite), Mn (1.5–3 wt%, causing solid-solution strengthening in ferrite), Cr (to retard pearlite or bainite formation), and Si (to enhance ferrite transformation) [5]. Although more than 50 years have passed since it was invented, researchers are still trying to optimise its chemical composition and thermomechanical processes to achieve better mechanical properties and push its performance to the top-right corner of the ‘Banana (steel strength ductility) diagram’.

Rapid alloy prototyping (RAP) provides an approach to developing and characterising new alloy variants without the need to produce large quantities of material. Recently, big progress has been reported to further increase the RAP speed by reducing the material to 20–140 gram casts per composition with a target of 100 compositions being made and tested per week [6]. The 20–140 grams of the material requires the re-design of a miniaturised specimen as the material is not enough to manufacture the standard bars, as shown in Figure 1.

The RAP strips were produced by the authors’ group. To investigate the validities of the outcomes obtained from the designed miniaturised non-standard bars, tensile tests were conducted. The yield strength, tensile strength, uniform elongation, and total/fracture elongation can be compared directly [8]. However, the post-necking behaviour usually cannot be easily observed. This stage is important as the degradation of the material can result in different necking and failure modes. Therefore, the influence of the size effect on the post-necking behaviour has attracted the researchers’ attention.

The size effect is usually represented through the aspect ratio (AR), which is the ratio of the tensile specimen’s parallel width to its parallel thickness in most of the studies. On the contrary, some researchers have defined the AR the other way around: the ratio of its thickness to its width [9]. Kohyama et al. studied the size effects (AR) on the tensile properties of austenitic stainless steel experimentally. Two types of bars, with parallel width values of 1.2 and 2.4 mm, were manufactured. Different AR values were achieved by varying the bars’ thickness. The results showed that the specimens’ fracture angles vary from 55° to 90° as the AR varies from 0.1 to 0.8 [9]. Tvergaard investigated the necking modes with different AR values through a full three-dimensional numerical analysis. This work focused on the influence of the AR on the necking mode transition: diffuse necking and localised necking [10]. The results indicated that the critical AR values for the necking mode transition are between three and six. It is worth noting that, even for the large AR dimensions, which are present in the case that localised necking is dominant, the diffuse necking may still occur just after the stress reaches the tensile strength. It is hard to observe as the time window is small. McClintock and Zheng studied the strips’ fracture angles by using cold-finished steel. The results suggested that 55° and 53° are the fracture angles for isotropic and anisotropic materials [11]. Mikkelsen developed a non-local continuum model, which made the post-necking behaviour become mesh independent. The relationship between the AR and the necking mode (diffuse necking and localised oblique necking) transitions was discussed [12]. Later, Mikkelsen studied the post-necking behaviour for a uniaxial tensile test specimen with a rectangular cross-section based on the developed approximate enhanced 2-D plane stress finite element model [13]. This newly developed model has computational advantages compared to the traditional 3D models. Mikkelsen compared the predicted results from the model to the previous ones, as well as to the research, and they matched: the oblique localised necking mode, the two crossed localised necking modes, and the diffuse necking mode occur at ARs = 16, 10, and 7, respectively. Kwame studied the failure behaviour of pure titanium through the tensile test (the tensile bars’ thickness is 1.6 mm) [14]. The results indicated that the shear stresses are acting optimally when they are acting along a 45° to the fracture line and are concentrated in the localised necked region before failure.

However, there is little research about the size effect on DP steel’s post-necking behaviour, i.e., the necking mode transitions or the stress triaxiality evolution. More importantly, determining whether the designed miniaturised non-standard tensile bars used for RAP are representative enough to predict the mechanical properties through the large tensile specimens, especially for the post-necking behaviour, is also the central question of this work. This work is organised as follows: the numerical setup is introduced in Section 2, which includes the selected materials, the simulation domain, the solver, the mesh sensitivity test, and the boundary conditions. The experimental setup is presented in Section 3. The experiment and the modelling results are discussed in Section 4.1 and Section 4.2. We finish with some concluding remarks.

## 2. Numerical Setup

### 2.1. Materials

Dual-phase 800 (DP800) steel was selected due to its wide application in the automotive industry, as discussed in Section 1. The dual phases refer to the ferrite and martensite and the typical microstructure, as shown in Figure 2.

The phase fraction ratio of the ferrite to the martensite was determined using image analysis of multiple SEM images. The darker ferrite and the lighter martensite regions were first thresholded using ‘Image J’ (version 1.53e) analysis software, and the areas of each were measured after performing an image calibration step. The ratio of the martensite phase to the ferrite phase is about 3:7 for DP800 steel. The final rolled strip material was supplied by Tata Steel. The chemical composition of DP800 is shown in Table 1.

### 2.2. Geometries

Five tensile specimen geometries were selected: three of them with dimensions based on the International Organization for Standardization (ISO)’s standards of ISO 6892-1 [7] (with gauge lengths of 80 and 50 mm: hereafter denoted as A80 and A50, respectively) and standard ASTM E8/E8M from the American Society for Testing and Materials (ASTM) [15] (with a gauge length of 25 mm, hereafter denoted as ASTM25). The other two geometries, named Mini1 and Mini2, are designed miniaturised non-standard specimens, which will be used to test small amounts of the materials obtained from the RAP routine. The gauge lengths for Mini1 and Mini2 are 10 and 5 mm, respectively. These five tensile test specimens’ dimensions are identical to our previous research [8]. All the specimens have the same thickness (1.2 mm) as they were manufactured from a final rolled strip from the plant. To save time on computation, only the gauge length region was simulated, as performed by the previous researcher [16]. The aspect ratio (AR) is defined as the ratio of the parallel width to the parallel thickness. Detailed information related to the dimensions (simulation domains) and the corresponding AR has been summarised in Table 2. The drawings and photos of tensile specimens are showed in Figure 3.

### 2.3. Numerical Setup

The elastic properties are defined through Young’s modulus (207 GPa) and Poisson’s ratio (0.29). The plastic properties were defined through the proof stress and its corresponding plastic strain. Both the elastic and plastic properties were calculated through the benchmark uniaxial tensile tests. In this study, the stress strain data were estimated through Johnson–Cook method [17]:(1)$$\sigma =\left[A+B\epsilon^{n}\right]\left[1+Cln{\dot{\epsilon }}^{*}\right]\left[1-T^{*m}\right],$$
where σ and ϵ are the proof stress and the equivalent plastic strain, and ${\dot{\epsilon }}^{*}=\dot{\epsilon }/{\dot{\epsilon }}_{0}$ is the dimensionless plastic strain rate for ${\dot{\epsilon }}_{0}$ = 1.0 s${}^{-1}$. $T^{*}$ is the homologous temperature. A,B,n,C, and *m* are material constants. For this work, the tests were conducted at room temperature, and, according to previous research [8,18], DP800 steel is not a strain rate-sensitive material within the strain rate range of the current experiment. Therefore, Equation (1) can be modified as follows:(2)$$\sigma =A+B\epsilon^{n},$$

The anisotropic behaviour was determined by the Hill48 model [19].

The Hill48 model uses the yield stress ratio ($R_{ij}$) to define yield behaviour. For a thin plate, the $R_{ij}$ values can be calculated through the following equations:(3)$$R_{11}=1,$$
(4)$$R_{22}=\frac{1}{\sqrt{\frac{r_{0}}{r_{90}\left(1+r_{0}\right)}+\frac{r_{0}}{\left(1+r_{0}\right)}}},$$
(5)$$R_{33}=\frac{1}{\sqrt{\frac{r_{0}}{r_{90}\left(1+r_{0}\right)}+\frac{1}{\left(1+r_{0}\right)}}},$$
(6)$$R_{12}=\sqrt{\frac{3\left(r_{0}+1\right)r_{90}}{\left(2r_{45}+1\right)\left(r_{0}+r_{90}\right)}},$$
(7)$$R_{13}=1,$$
and
(8)$$R_{23}=1,$$
where $r_{0},r_{45}$, and $r_{90}$ denote the R-values at 0, 45, and 90 degrees of the rolling direction. Determination of the DP800 R-values is also one of the aims of this study, as the previous research on DP800 R-values is limited, which may be because DP800 is not a ‘traditionally’ forming steel/alloy. The results of this part of our work are discussed in Section 4.1.

The ductile damage model was adopted to model the fracture behaviour. The detailed theory of the ductile damage model can be referred to in the Abaqus manual [20]. Briefly, for the ductile damage model, both the damage initiation and the evolution need to be defined. The damage initiation was defined through the fracture stain, the triaxiality, and the strain rate. The criterion for damage initiation is met when the following condition is satisfied:(9)$$w_{D}=\int \frac{d{\bar{\epsilon }}^{pl}}{{\bar{\epsilon }}_{D}^{pl}\left(\eta ,{\dot{\bar{\epsilon }}}^{pl}\right)}=1,$$
where $w_{D}$ is a state variable that increases monotonically with plastic deformation. ${\bar{\epsilon }}^{pl}$, ${\bar{\epsilon }}_{D}^{pl}$, η, and ${\dot{\bar{\epsilon }}}^{pl}$ denote the equivalent plastic strain, the equivalent plastic strain at the onset of damage, the stress triaxiality, and the equivalent plastic strain rate. For the damage evolution, the displacement at failure, ${\bar{u}}_{f}^{pl}$, needs to be defined. Due to the mesh sensitivity around elements with reduced stiffness, a stress-displacement response is used [21]. During the damage evolution, the effective plastic displacement, ${\bar{u}}^{pl}$, can be defined through the following equation:(10)$${\bar{u}}^{pl}=L{\bar{\epsilon }}^{pl},$$
where *L* is the characteristic length of the element. In this work, a linear evolution of the damage has been used. After ${\bar{u}}_{f}^{pl}$ is defined, the damage variable increases according to:(11)$$d=\frac{{\bar{u}}^{pl}}{{\bar{u}}_{f}^{pl}}.$$
This indicates that when the effective plastic displacement reaches the value of ${\bar{u}}_{f}^{pl}$, the material stiffness will be fully degraded (d=1). Inverse modelling was conducted to determine the damage initiation (the values of fracture stain, the triaxiality, and the strain rate) and evolution (the value of displacement at failure) parameters. The results of this part of the work are discussed in Section 4.2.1.

The boundary conditions were defined as follows: one end of the GL is fixed, and a displacement was applied through a reference point, which is coupled with the other end of the GL. The numerical simulations were conducted by using the Abaqus/Explicit solver. It is worth pointing out that a new advance in modelling the fracture of viscoelastic solids has been presented recently by Thamburaja et al. [22] and Sarah et al. [23]. Compared to the local damage criterion, the proposed non-local damage criterion showed its advantage: the results are independent of the mesh density, element type, and orientation. The flow curves, the force-displacement curves, and the crack propagation were also well predicted. Additionally, mesh-independent methods, such as the extended finite element method (XFEM), for example, were also discussed and well-reviewed by the researcher, such as the work completed by Rege and Lemu [24]. One of the main objectives of this work is to investigate the fracture angle variation, which depends mainly on the stress distributions. Therefore, a traditional mesh sensitivity test was conducted because the post-necking behaviour is sensitive to meshes in the necked region. Because it is hard to predict the necking/fracture region, the mesh was refined all along the PW (*x*-axis), GL (*y*-axis), and THK (*z*-axis) directions. Since the damage feature is not symmetrical, the full gauge length is adopted. Four meshes were selected, and the details are presented in Table 3.

Mesh 3 was selected, ensuring good precision at a reasonable computational cost. Figure 4 shows the mesh details and the coordinate system.

Table 4 shows the detailed dx,dy, and dz values of Mesh 3 for different dimensions of the specimens.

## 3. Experiment Setup

In order to determine the yield stress ratios ($R_{i,j}$), which are used to define the anisotropic behaviour during the simulations, tensile tests were conducted to determine the plastic strain ratios (R-value) of DP800. Three standard tensile bars were adopted: A80, A50, and ASTM25, respectively. The Tinius Olsen H25KS tensile machine (with a maximum applied force of 25 kN) was used to record the applied forces and to calculate stresses later on, and a video extensometer (XSight 9MPX) was used to capture strains during the test (Figure 5).

The video extensometer records the pictures with a pre-set frequency. A post-processing step is required to allow the program to calculate the strain distribution of each frame whilst conducting digital image correlation (DIC) analysis. The strain rate followed the standard [7], and the strain rates for Range 2 and 4 are 2.5 $\times \;10^{-4}$ L/s and 6.7 $\times \;10^{-3}$ L/s. Range 2 is defined as the range in the plastic region from the starting yield strength to the post-yield strength (which is a lower yield strength). Range 4 is defined as the stage between the lower yield strength and the fracture point. Once the experiment has been completed, the R-values are calculated using the following equation:(12)$$r=\frac{ln(1+e_{W,Eng})}{ln\left(1+e_{W,Eng}\right)+ln\left(1+e_{l,Eng}\right)},$$
where $e_{W,Eng}$, and $e_{l,Eng}$ denote the strain along the parallel width and pulling directions.

## 4. Results and Discussion

### 4.1. Experimental Results

Table 5 shows the previous work (from 2007) compared to the DP800 R-values and the measured R-values of A80, A50, and ASTM25 in the current work.

The results show that the measured DP800 R-values are located within the range of the previous research. The measured R-value trends show that $r_{45}>r_{90}>r_{0}$, and this also agrees with the previous research. The reasons for this distribution were discussed by Tasan et al. [33]: that sharp deformation bands nucleate at ferrite grains and propagate in the softest route within the microstructure with angles of 45–50 to the loading direction. In this work, the average values of A80, A50, and ASTM25 for 0°, 45°, and 90° were used for calculate the yield stress ratios of $R_{ij}$ in Equations (4)–(7).

### 4.2. Modelling Results

#### 4.2.1. DP800 Damage Model Calibration

Since the full shape of a tensile curve is dependent on the elastic, plastic, and failure mechanisms, the tensile parameters were optimised using an inverse modelling approach, during which the model parameters for hardening and damage are design variables in the optimisation problem of minimising the error between the predicted tensile curve and those obtained from a typical A80 test piece. This optimisation was performed using a downhill simplex algorithm in the commercially available Isight software (version 2022), using Abaqus as the finite element solver.

This method allowed the parameters to be found such that the simulated tensile curve resembled the experimental curve for DP800, as shown in Figure 6.

However, upon close inspection, the onset of material failure immediately after the UTS does not correlate well, and the results show that, for DP800 steel, it is hard to match the uniform elongation point (UTS point), the stress strain damage curve, and the fracture strain at the same time. This is likely due to the fact that the damage is distributed less uniformly during tests, whereas the model assumes that damage is purely a function of plastic strain and homogenous across the gauge section. This would be particularly evident in dual-phase steel, such as DP800, in which failure occurs at the interface between the ferrite and the martensite, with the latter phase being far more brittle. Figure 6 shows the engineering stress strain curves’ comparison of the experiment (raw data in black) and the modelling (in red) of DP800 based on the A80 dimension. The lower figure shows that although the necking point of the modelling occurs later than that of the experiment (18.78% and 14.5% respectively), the stress increase is 0.5% for the modelling, and the decrease is 4.2% for the experiment during this period (a strain of two [14.5%, 18.78%]). It is believed that this stage has little effect on post-necking behaviour.

Further validation was conducted through comparisons of the digital image correlation (DIC) and the modelling results, as shown in Figure 7a–d and Figure 7e–h, respectively.

The plastic strain was measured and calculated in the vicinity of the necking zone of the tensile bar throughout the tensile test. The results show that, at first, the tensile bar was deformed uniformly until the stress reached the ultimate tensile strength (UTS). The strain then started to concentrate in the necking zone. As the test continued, the strain concentrated along a cross-shaped structure, as shown in Figure 7e,g. After this, one of the cross-shaped bands became saturated, and the plastic strain continued to concentrate along the other band, which triggered the final fracture of the bar, as shown in Figure 7d,h. The results increase confidence in the numerical simulation’s prediction.

#### 4.2.2. AR Effect: Necking and Fracture Strains

The engineering stress strain curves for A80 (AR= 16.7), A50 (AR= 10.4), ASTM25 (AR=5), Mini1 (AR= 2.5), and Mini2 (AR= 1.67) are shown in Figure 8.

The results show that the necking starts at a similar strain for the standard specimens where *AR* ∈ [5, 16.7]. As the AR value reduces below this level, the necking starts earlier; however, since the tensile curve is relatively flat in this region, the UTS remains similar, as shown in Figure 9a.

As the engineering strain increases beyond this point, the stress decreases most rapidly in the samples with the largest AR (A80 and A50). Therefore, although the onset of necking occurs earliest in the sample with the smallest AR, the total elongation is the greatest for this specimen, as shown in Figure 9b. This trend of the fracture strain increasing as the AR decreases can be explained by applying Oliver’s Law [34]. This law relates the total (or fracture) elongation to the slimness ratio and the ratio of the gauge length to the square root of the cross-section area of the gauge section, with larger elongation values occurring with small slimness ratios. Figure 8 shows that this increase in the total elongation is due to an increase in the post-necking strain, as opposed to the uniform elongation.

Table 6 shows the comparison between the simulation and the experiment of the uniform elongation ($e_{U}$) and the total elongation ($e_{f}$).

The results indicate that the modelling results correlate well with the experimental values for the total elongation for all of the specimens, except for Mini2. The hardening and damage parameters of the model were obtained by minimising the error between the experimental and predicted tensile curves for the A80 specimen, and, therefore, a good correlation is expected for this sample. The modelling shows that the material parameters predict the total elongation of the smaller specimens well, with the exception of Mini2, which has a much smaller AR (1.67 vs. 16.7). The reason for this is that the post-necking behaviour of the small specimen, in terms of the stress triaxiality, the accumulation of the plastic strain, and the fracture angle, is very different from the large specimen. This is discussed in more detail below. The discrepancy between the predicted and experimentally obtained uniform elongation is described in detail in Section 4.2.1 and is due to how the damage parameters are characterised.

#### 4.2.3. AR Effect: Stress Triaxiality Evolution

Stress triaxiality, η, is defined as the ratio of the mean (hydrostatic) stress to the effective stress and is an important parameter when considering a material fracture. The evolution of stress triaxiality along the most necked zone varies in each tensile specimen, as shown in Figure 10.

The simulation results show that, prior to necking, the triaxiality value remains a constant value of 1/3, as the specimens were under uniaxial tensile conditions. At the onset of necking, the triaxiality increases in all specimens. This is due to the necked region behaving as a small notch, which deepens as the test continues. The triaxiality initially increases at a lower strain for the smaller specimen than the larger standard specimens. This is due to the deformation occurring more readily on both the thickness and width directions in the smaller specimen because of the lower aspect ratio. For the larger specimens, the large width-to-thickness ratio creates plane strain conditions in the width direction in which little transverse strain can occur. As the samples continue to deform, this constraint will result in a higher hydrostatic stress (negative pressure) across the specimen’s width, which results in the triaxiality increasing more rapidly in the larger specimens. For the smaller samples with a low aspect ratio, deformation in both the thickness and width directions occurs more readily, resulting in a slower increase in the hydrostatic stress and, thus, a slower increase in the triaxiality. The differences in the deformation in the width and thickness directions during necking for both the A80 and Mini2 specimens are shown in Figure 11.

#### 4.2.4. AR Effect: Necking Modes

As previously mentioned, the aspect ratio of the specimen’s gauge section affects the stress distribution and, hence, the accumulation of plastic strain in the necked region. These differences can be highlighted by comparing the necked regions of the specimens with the largest and smallest aspect ratios: A80 and Mini2, respectively.

Figure 11a shows how the equivalent plastic strain (PEEQ) evolves from the onset of a neck through to the failure point, with intermediate regions at 50% of the post-necking period (PNP) strain and 75% of the PNP strain. Figure 11b,c show the isolines of the evolution of specimen displacement in the width direction (x,U1) and specimen displacement in the thickness direction (z,U3), respectively, over the same post-necking period.

For both samples, the PEEQ becomes non-homogenous at the onset of necking, with higher values occurring at the neck location. However, as deformation continues, the PEEQ concentrates in bands at approx. a 57° to the applied load in the larger specimen. These bands occur in both orientations forming a cross-shaped structure. As deformation continues further, the strain at one of these bands becomes dominant, and eventually, failure occurs at this angle to the applied load. In this case, localised necking is dominant, resulting in a shear-type fracture. For the smaller tensile specimen, with an AR = 1.67, these distinct deformation bands do not form during necking, and failure occurs perpendicular (90°) to the loading direction. For this specimen, the necking is diffuse as deformation can occur more readily in both the width and thickness directions, and fracture is more ductile.

#### 4.2.5. Fracture Angle

The fracture angle is defined as the angle between the loading and parallel width directions. Figure 12 shows both the predicted and typical experimentally obtained fracture angles for the DP800 tensile specimens with different AR values.

For both the experimental and modelling results, changes in the fracture angle were observed as the aspect ratio of the gauge length changed, with fracture angles close to 55° for specimens with a large AR and 90° for specimens with a low AR. Multiple tensile tests were performed, resulting in a measure of the scatter for the fracture angle of the experimental samples. For intermediate AR values, the fractures consisted of regions of failure at a 90° to the loading direction and regions at an angle of ∼55°. This can be seen in Figure 12.

Both the predicted and experimental fractures for this study and for previous studies on different alloys are shown in Figure 13.

For this figure, an average fracture angle is used for which the failure mode consists of regions of different fracture angles, such as those observed for the A50 tensile specimen. The range of different fracture angles observed for equivalent tensile specimens can be seen in Figure 14.

This range of results explains the high standard deviation obtained for the experimentally observed fracture angle for the A50 and A80 tensile specimens. This variability may be due to the inhomogeneity of fracture, whereby the final fracture will be dependent on the location of crack initiation. Additionally, the morphology of the dual-phase microstructure may influence the location of crack initiation and, hence, the final fracture. Finally, although the tensile specimens are aligned carefully in the tensile test machine, slight variabilities in alignment may exist, which may affect fracture. For the specimens with a low AR (ASTM25, Mini1, and Mini2), the fracture angles were consistently at a 90° to the loading direction.

The experimental results indicate that the critical AR at which the mode of fracture changes from a perpendicular fracture to an angled one, is approximately five. This value agrees with the analytical result obtained by Hill [37]. For the modelling work, the critical value of the AR falls in the range of 2.5–5. It is also observed that the transition in the fracture angle for DP800 is sharp compared to the other steels, such as 10Cr-2Mo. This is likely due to the relatively high strength of the dual-phase steel.

## 5. Conclusions and Future Work

This work aimed to investigate the tensile specimen size effect on the post-necking behaviour of dual-phase 800 steel for standard tensile specimens down to miniaturised non-standard specimens through both experiment and modelling. The main conclusions are summarised as follows:1.The variation in the R-value with the orientation to the loading direction followed a similar trend as in previous studies: $r_{45}>r_{90}>r_{0}$ for all standard tensile bars. Although there was some variation between the different specimen sizes, no clear trend was observed, and the differences were minimal. This is likely due to slight differences in the microstructure as a result of different thermomechanical processing.2.A good correlation could be found between the predicted and experimentally observed UTS and total elongation; however, the drop in stress during the post-necking period was difficult to capture. This is likely due to the progressive failure of the material, which is not well captured by the model.3.The stress triaxiality at the notch increased earliest for the specimens with the lowest AR. The triaxiality increased later for the specimens with a larger aspect ratio (the ratio of parallel width to thickness, AR) value, but this triaxiality then increased more rapidly. This is due to the higher constraint of strain in the width direction for larger ARs, which results in a higher hydrostatic stress and, hence, a higher triaxiality.4.The fracture angle and dominant necking modes are dependent on the AR. The standard A80 test piece exhibited a fracture angle of $∼55^{o}$ (localised necking dominant), whereas smaller specimens with a low AR exhibited fractures perpendicular to the loading direction (diffused necking dominant).

The results indicate that care must be taken when comparing the post-necking behaviour of small-scale tensile tests, such as those completed as part of a rapid alloy prototyping program, to those of standard full-size test specimens, as post-necking behaviour is highly dependent on test piece’s AR values. The current model developed more confidence in understanding the post-necking behaviour of small-scale tensile tests. Future work may focus on comparing the predicted results, i.e., fracture angles, through different damage models, such as nonlocal damage criterion.

## Figures and Tables

**Figure 1 materials-16-01458-f001:**
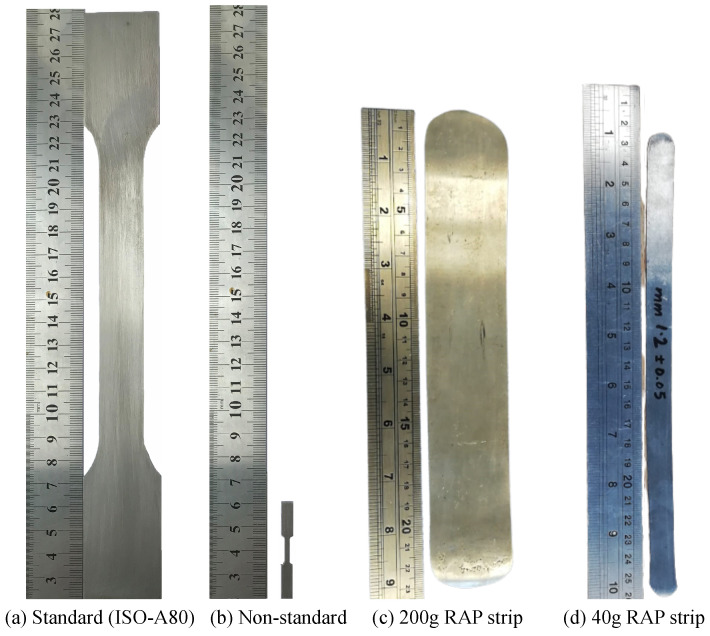
(**a**) Standard [7] and (**b**) non-standard [8] tensile specimens and the strips (**c**,**d**) obtained from the rapid alloy prototyping routine. The miniaturised non-standard tensile specimen is a better choice due to the limited amount of rapid alloy prototyping materials.

**Figure 2 materials-16-01458-f002:**
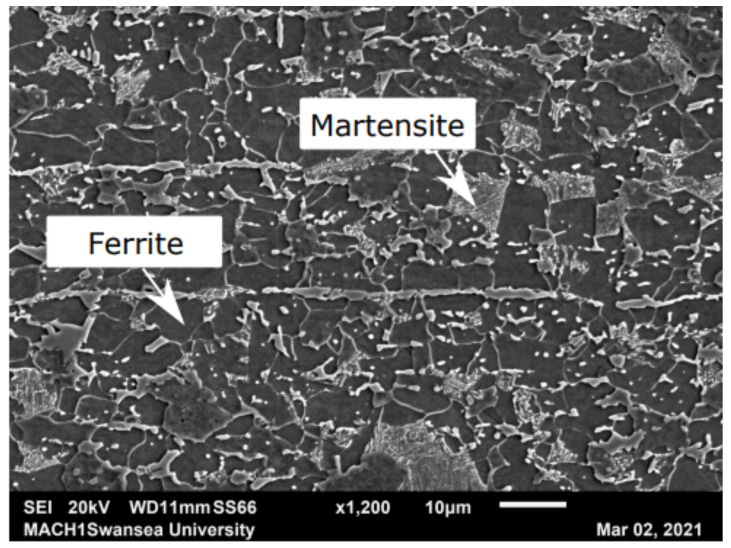
Typical microstructure of DP800 steel. The ratio of martensite phase to ferrite phase is about 3:7 for DP800 steel.

**Figure 3 materials-16-01458-f003:**
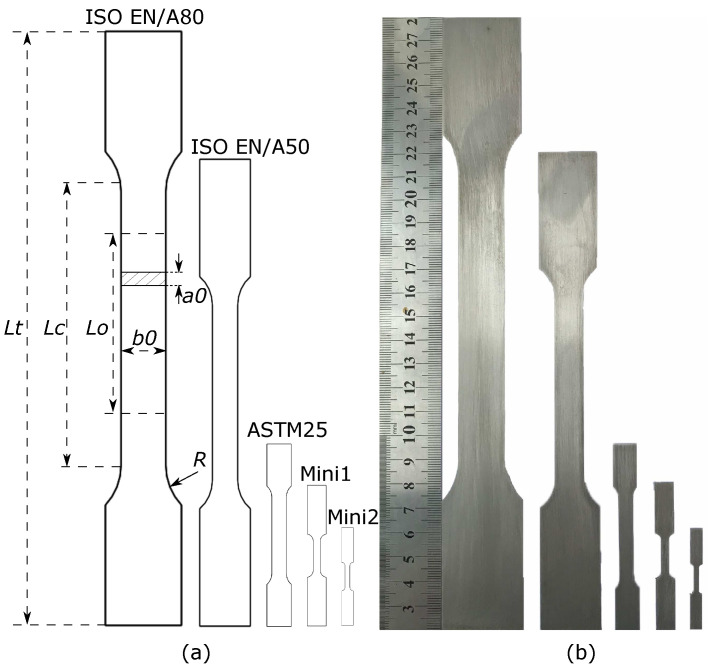
CAD drawings (**a**) and photos (**b**) of different tensile test specimens. Lt,Lc,Lo,b0,R, and a0 denote the total length, the parallel section length, the gauge length, the original width, the shoulder radius, and the original thickness of the test piece, respectively.

**Figure 4 materials-16-01458-f004:**
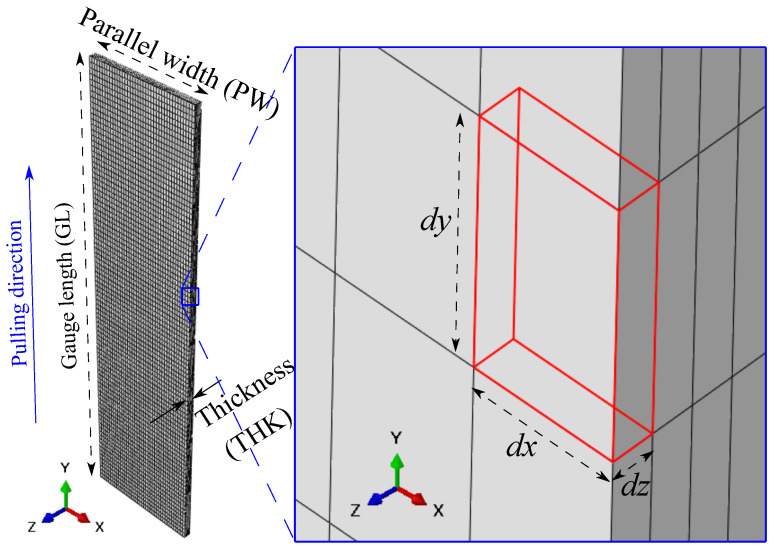
Mesh of the computation domain and definitions of the coordinate system, PW, GL, THK, dx, dy, and dz.

**Figure 5 materials-16-01458-f005:**
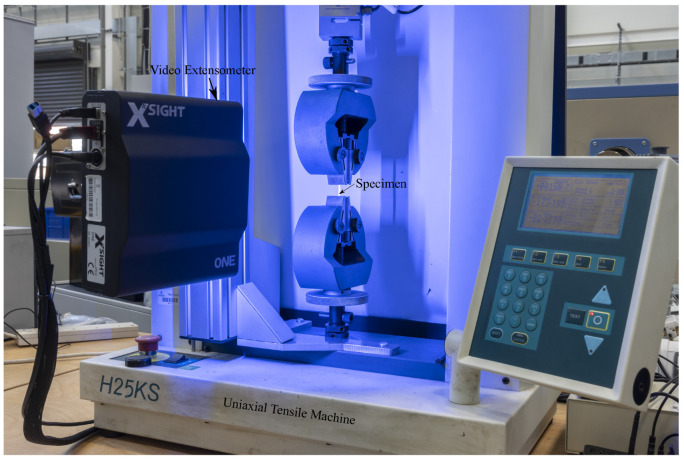
Tinius H25KS tensile machine and XSight 9MPX video extensometer.

**Figure 6 materials-16-01458-f006:**
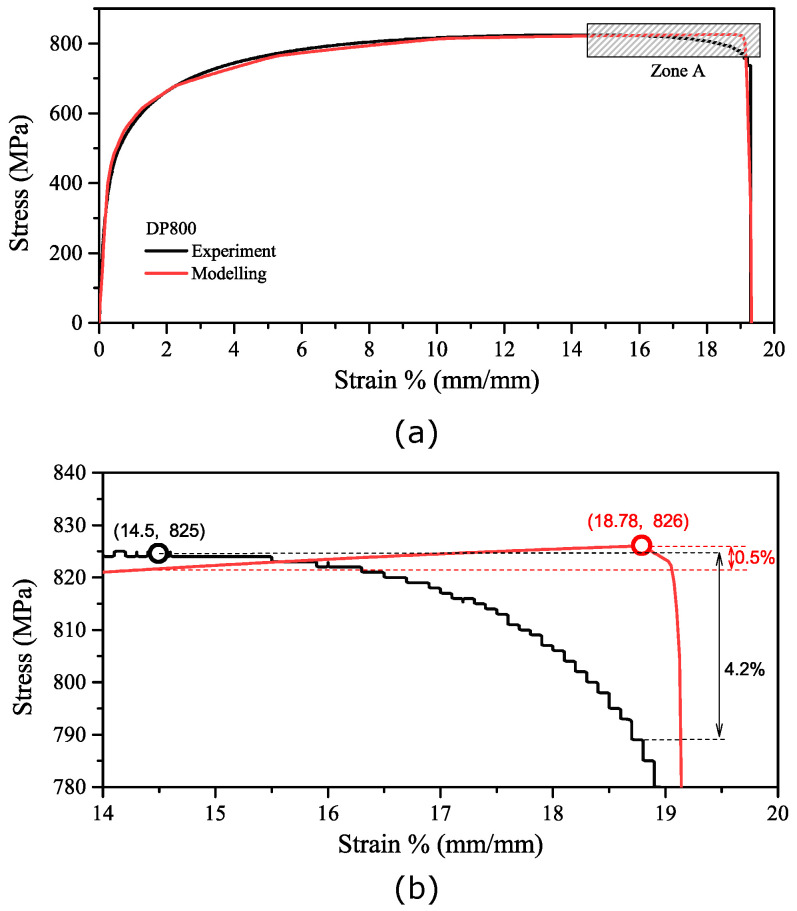
The engineering stress strain curves comparison between the experiment (raw data in black) and modelling (in red) of DP800 based on A80 dimension; (**a**): the whole range of the stress strain curve and (**b**): the enlargement of Zone A.

**Figure 7 materials-16-01458-f007:**
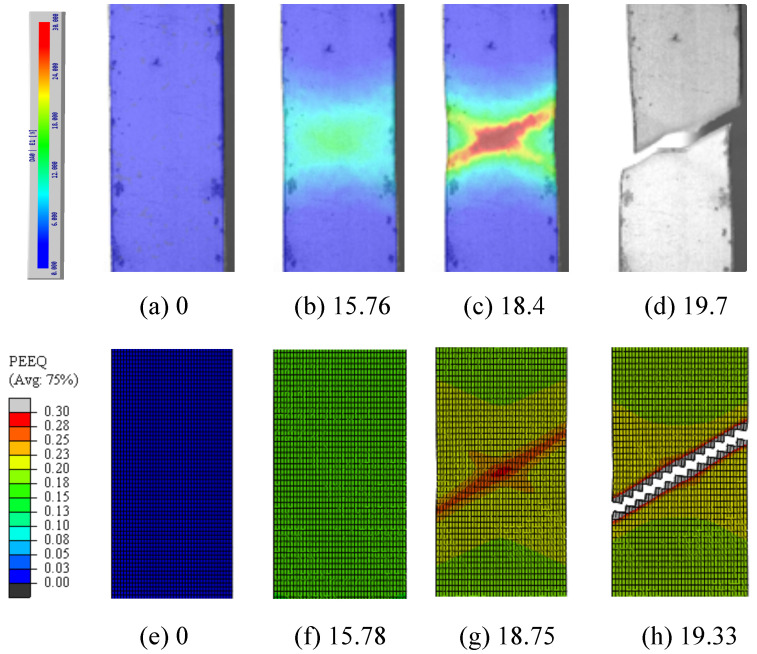
The plastic strain distribution comparisons between the digital image correlation (**a**–**d**) and modelling (**e**–**h**) of frames with different engineering strains based on A80 specimen. Both plastic strain distribution and magnitude are matched.

**Figure 8 materials-16-01458-f008:**
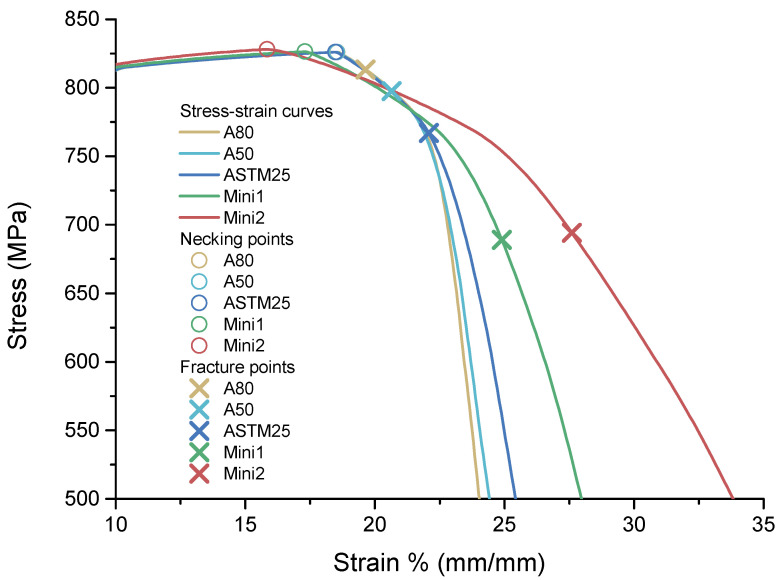
The engineering stress strain curves comparison for A80 (*AR* = 16.7), A50 (*AR* = 10.4), ASTM25 (*AR* = 5), Mini1 (*AR* = 2.5), and Mini2 (*AR* = 1.67).

**Figure 9 materials-16-01458-f009:**
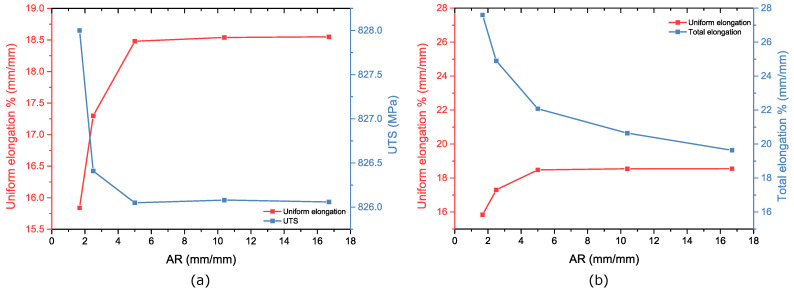
(**a**): AR effect on the uniform elongation and UTS. (**b**): AR effect on the uniform and total elongations.

**Figure 10 materials-16-01458-f010:**
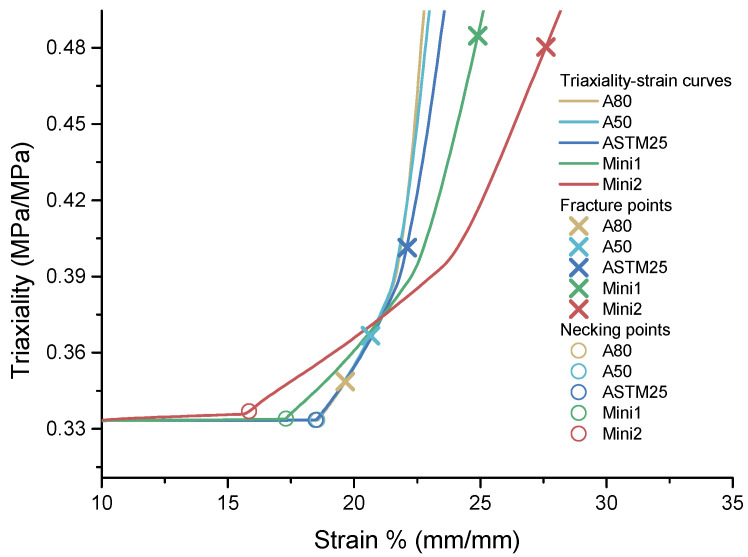
The stress triaxiality evolution for A80 (*AR* = 16.7), A50 (*AR* = 10.4), ASTM25 (*AR* = 5), Mini1 (*AR* = 2.5), and Mini2 (*AR* = 1.67).

**Figure 11 materials-16-01458-f011:**
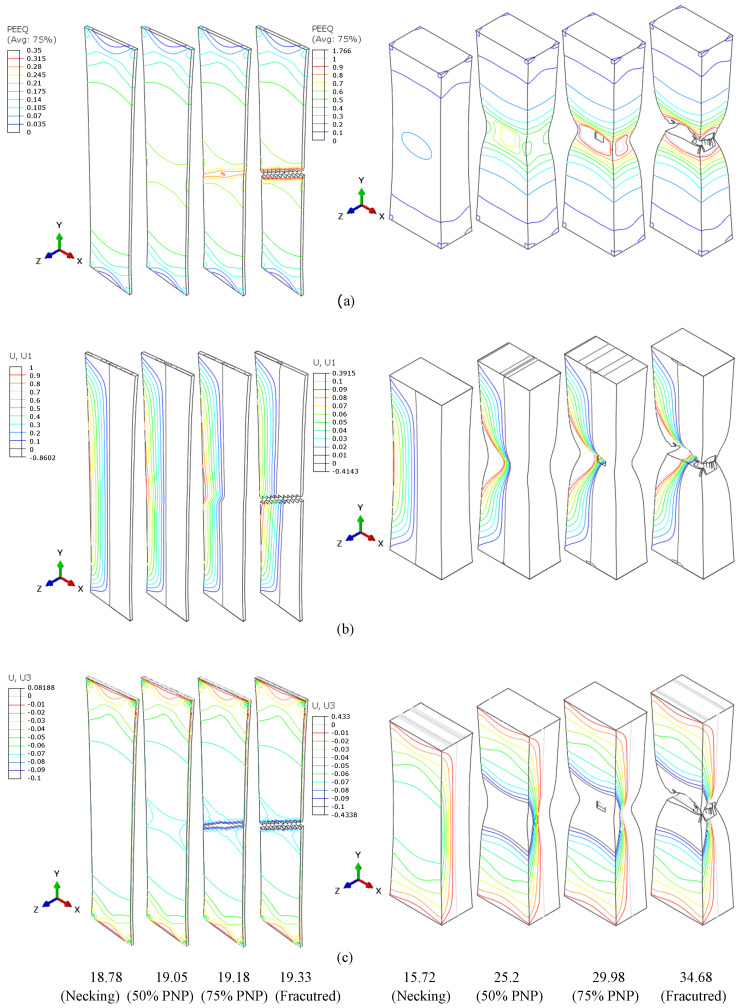
The evolutions of the equivalent plastic strain (PEEQ) (**a**), the *x* (U1) displacement (**b**), and the *z* (U3) displacement (**c**) for ARs = 16.7 (A80) and =1.67 (Mini2).

**Figure 12 materials-16-01458-f012:**
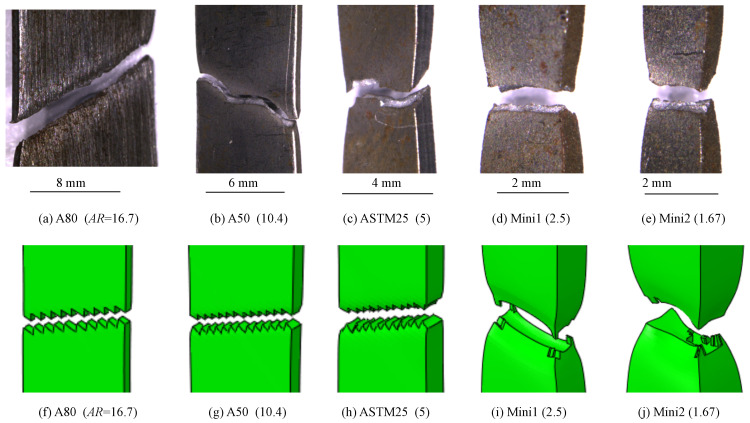
The fracture surfaces of DP800 tensile specimens with different AR values. The fracture angle (the acute angle between the pulling direction and the fracture surface plane) increases as the AR value is decreased; (**a**–**e**): experiment and (**f**–**j**): modelling.

**Figure 13 materials-16-01458-f013:**
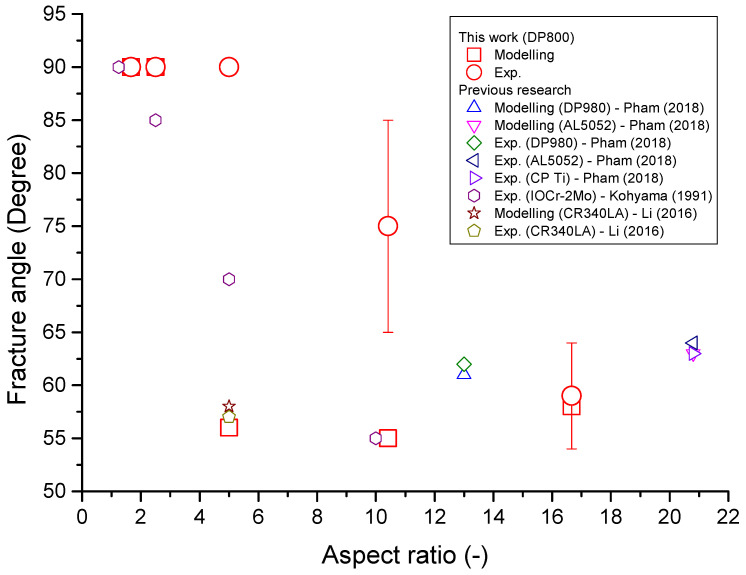
Fracture angle vs. aspect ratio for different steels. For DP800, the transition AR values are located between 10.4 and 5 for the experiment and 5 and 2.5 for the modelling. In the legend, Pham (2018), Kohyama (1991), and Li (2016) are the references of [9,35,36], respectively.

**Figure 14 materials-16-01458-f014:**
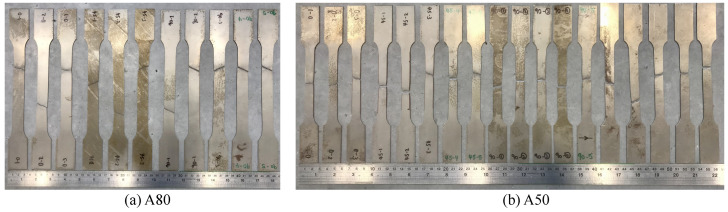
Broken A80 (**a**) and A50 (**b**) tensile specimens’ appearances. The fracture angles of the A50 bars present a large deviation.

**Table 1 materials-16-01458-t001:** Chemical compositions for dual-phase 800 steel (% weight). The data were given by Tata Steel.

C	Si	Mn	P	S	Ni	Cu	Cr
0.136	0.249	1.77	0.011	0.0027	0.018	0.024	0.558

**Table 2 materials-16-01458-t002:** Tested tensile bar dimensions. $L_{t},L_{c},L_{o},b_{0},R$, and $a_{0}$ denote the total length of the test piece, the parallel length, the gauge length, the original width of the parallel length of a flat test piece, the shoulder radius, and the thickness of the bar, respectively. The slimness ratio (*K*) and the aspect ratio (AR) are defined as $\frac{L_{o}}{\sqrt{a_{0}\cdot b_{0}}}$ and $\frac{b_{0}}{a_{0}}$.

	$L_{t}$	$L_{c}$	$L_{o}$	$b_{0}$	*R*	$L_{o}/L_{c}$	$L_{o}/b_{0}$	($L_{c}$−2$b_{0}$)/$L_{0}$	$a_{0}$	*K*	AR	No. of Sample
	(mm)	(mm)	(mm)	(mm)	(n/a)	(n/a)	(n/a)	(n/a)	(mm)	(n/a)	(n/a)	(n/a)
A80	260	120	80	20	25	0.67	4	1	1.2	16.33	16.7	6
A50	200	75	50	12.5	15	0.67	4	1	1.2	12.91	10.4	6
ASTM25	76	32	25	6	6	0.78	4.17	0.8	1.2	9.32	5	6
Mini1	60	12.5	10	3	3	0.8	3.33	0.65	1.2	5.27	2.5	6
Mini2	41	9	5	1.5	2	0.56	2.5	1	1.2	3.23	1.67	6

**Table 3 materials-16-01458-t003:** Details of the four selected meshes. PW, GL, and THK stand for parallel width, gauge length, and thickness of the tensile specimen, respectively.

	Mesh 1	Mesh 2	Mesh 3	Mesh 4	Increasing Ratio
No. div. of PW (-)	11	21	**41**	81	∼2
No. div. of GL (-)	31	61	**121**	241	∼2
No. div. of THK (-)	3	5	**9**	17	∼2
Total element. No. (-)	1023	6405	**44,649**	331,857	-
Total node. No. (-)	1536	8184	**51,240**	357,192	-
Error of necking strain (-)	1.87 × 10${}^{-2}$	1.07 × 10${}^{-2}$	**8.02 × 10${}^{-3}$**	0	-
Error of stress at necking strain (-)	4.22 × 10${}^{-5}$	3.15 × 10${}^{-6}$	**2.73 × 10${}^{-6}$**	0	-

**Table 4 materials-16-01458-t004:** In detail, specimen’s dimensions and element edges for Mesh 3.

	PW	*dx*	GL	*dy*	THK	*dz*
	(mm)	(mm)	(mm)	(mm)	(mm)	(mm)
A80	20	0.488	80	0.661	1.2	0.133
A50	12.5	0.305	50	0.413	1.2	0.133
ASTM25	6	0.146	25	0.207	1.2	0.133
Mini1	3	0.073	10	0.083	1.2	0.133
Mini2	2	0.049	5	0.041	1.2	0.133

**Table 5 materials-16-01458-t005:** Research compared to DP800 R-value measurement from 2007.

Lead Author	Ref	Dimension	THK (mm)	r_0_	r_45_	r_90_
Walp	[25]	ISO-A80	1.5	0.63	-	-
Cardoso	[26]	∼ISO-A50	1.2	0.579	1.077	0.696
Cardoso	[27]	∼ISO-A50	1.2	0.516	1.237	0.711
Beres	[28]	Nakajima	1	0.65	0.77	0.72
Zaman	[29]	Gauge length 42 cm	1	0.7	0.97	0.82
Kim	[30]	Gauge length 42 cm	1	0.7	0.97	0.82
Almeida	[31]	∼ASTM25	-	0.955	0.978	0.897
Unlu	[32]	-	-	0.71	0.88	0.83
**Current work**	-	ISO-A80	1.2	0.74	1.32	1.01
**Current work**	-	ISO-A50	1.2	0.75	1.14	1.03
**Current work**	-	ASTM25	1.2	0.97	1.23	0.93

**Table 6 materials-16-01458-t006:** The comparison between the simulation and the experiment of the uniform elongation ($e_{U}$) and the total elongation ($e_{f}$). UB and LB stand for upper and lower bounds, respectively.

	$e_{U}^{\mathrm{FE}}$	Mean $e_{U}^{\mathrm{EXP}}$	Exp. Bounds	Mean Discrepancy	Bounds Discrepancy	$e_{f}^{\mathrm{FE}}$	Mean $e_{f}^{\mathrm{EXP}}$	Exp. Bounds	Mean Discrepancy	Bounds Discrepancy
	(n/a, %)	(n/a, %)	(n/a, %)	(n/a, %)	(n/a, %)	(n/a, %)	(n/a, %)	(n/a, %)	(n/a, %)	(n/a, %)
A80	18.78	14.4	UB: 15.5	23.3	UB: 17.47	19.33	19.7	UB: 19.8	1.94	UB: 0.51
			LB: 13.6		LB: 27.58			LB: 17.6		LB: 10.66
A50	18.72	14.5	UB: 14.7	22.5	UB: 21.47	19.68	20.5	UB: 20.7	4.17	UB: 0.98
			LB: 13.9		LB: 25.75			LB: 17.0		LB: 17.07
ASTM25	18.72	15.1	UB: 15.9	19.3	UB: 15.05	20.72	22.7	UB: 23.4	9.56	UB: 12.93
			LB: 14.1		LB: 24.68			LB: 21.3		LB: 2.8
Mini1	17.4	15.3	UB: 15.9	12.1	UB:10.92	26.9	25.8	UB: 26.5	4.09	UB: 2.71
			LB: 14.9		LB: 14.37			LB: 22.4		LB: 13.18
Mini2	15.72	13.8	UB: 14.1	12.2	UB: 10.31	34.68	27.4	UB: 30.5	20.99	UB: 12.05
			LB: 12.5		LB: 20.48			LB: 25.4		LB: 29.35

## Data Availability

Not applicable.

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
