# Peer review of "Size Effect on the Post-Necking Behaviour of Dual-Phase 800 Steel: Modelling and Experiment"

_materials, 2023, doi:10.3390/ma16041458_

Round 1
Reviewer 1 Report
The comments for manuscript authors is in the attached file.

Reviewer 2 Report
In the abstract, what is A80 and RAP? It should be explained or given in long form for the first time.
In the introduction section, the general information given for dual-phase steels in the first part is insufficient. This type of steel has widespread application, especially in the automotive industry. In addition, it increases passenger safety with its high strength in thin sections and contributes positively to environmental factors such as weight, fuel consumption, and carbon footprint thanks to its lightness. Therefore, this part needs to be elaborated.
In the continuation of the introduction, several previous studies in the literature are included. A literature review should be developed. In addition, the evaluation of the studies in the literature and the novelty content and differences of this study should be stated.
For the questions given in the introduction part, the introduction part should be created by explaining them in paragraphs, not in question format.
In Part 2, the phase volume ratio for DP800 steel is given. How was this calculated? Specified reference should be given if provided from the literature or by the manufacturer.
How was the chemical composition determined?
An exemplary tensile test specimen drawing should be placed. On this drawing, the required parts or the gauge length, etc. varying according to the sample size should be shown.
The parts given in Table 2 can be given on this drawing.
It is stated in Equation 1 that there is a JC model, but this is only the first part of the JC plasticity model. There are also C and m, that is, speed and temperature effect values in the model.
If different strain rate values are used, your analyzes will not be accurate because the C parameter is not used in your JC plasticity model.
Your article includes the sentence "The ductile damage model was adopted to model fracture behavior". However, the JC damage model is given. Only part of the plasticity model is available. If JC damage model is used, D1, D2, D3, D4 parameters and this model should be given.
Tensile curves for tensile tests performed at different sample sizes or AR and graphs comparing parameters such as the hardening coefficient obtained from these curves should be added. In addition, the fracture surfaces obtained as a result of the tensile tests performed at different sample sizes should also be examined.
The conclusion part is very long. In this section, the results obtained in the article should be expressed clearly and precisely as articles.
Reviewer 3 Report
This paper provides the combination of a theoretical, computational and experimental investigation of the sample size effect on the post-necking and fracture response of a dual-phase steel sample. It is definitely a topic worth investigating since the prediction of the fracture response and reliability of such an important construction material is of technological interest. The computational studies were performed using the FEM method (in particular, the ABAQUS FEA program). Results from the FEM are compared to experimental data, and the results seem acceptable. In summary, the results displayed in this paper are very useful to other researchers who are interested in this field.
However, there are some major points that must be addressed before this paper can be accepted for publication in a top journal like Materials, and they are as follows:
(1) The details of the constitutive theory are missing. For example, what is the stress-strain law employed? The plastic flow rule? What plasticity model is used ? Is the plastic volumetric contribution considered ? Is the plasticity treated as rate-independent or rate-dependent ?
(2) In Page 9, it is stated that "For this work, a linear evolution of damage with u^{pl} was used with eventual element failure, and element deletion,
occurring when \bar\epsilon^{pl} exceeds the plastic displacement at failure, u_f^{pl}". This statement resembles a fracture criterion, but it is misleading since strain is dimensionless whereas the plastic displacement has dimensions of length. This statement must be modified to reflect the actual fracture criterion.
(3) In Figure 6, it looks like the failed elements are focused in the region spanning one or two elements. This is very characteristic of a local-based fracture criterion. Hence, the simulated fracture response (that is, post-necking response) will be mesh sensitive, but this is somehow mitigated by the introduction of the L parameter (that is, the characteristic length of the element) in the constitutive model. However, this is a numerical fix and the authors must acknowledge this issue. The parameter L cannot be introduced at the constitutive level, and equation 9 is a numerical fix for the mesh sensitive response exhibited by the fracture response.
(4) With regards to simulating fracture using an element failure (or deletion) approach to obtain mesh insensitive response for the fracture process within an ABAQUS FEA framework, this paper has not performed a literature survey which is reflective of the current work in the area. For example, Reddy and co-workers [1,2] have shown within the ABAQUS FEA framework, the element failure/deletion method can be used to obtain mesh density independent and element type independent stress-strain responses and cracking patterns provided a non-local fracture criterion is used. The authors could use Reddy's [1,2] framework in their future works (since the authors are also using ABAQUS) and introduce a non-local fracture criterion to obtain mesh independent fracture responses without introducing a numerical fix at the constitutive level.
[1] Thamburaja et al. (2019) Computer Methods in Applied Mechanics and Engineering, vol. 354, pp. 871-903.
[2] Sarah et al. (2020) Mechanics of Advanced Materials and Structures, vol. 27, pp. 1085-1097.
If the above-mentioned points are satisfactorily addressed by the authors, this paper can be accepted for publication in Materials.
Reviewer 4 Report
1/ Please provide all geometrical values for the samples used in the tensile test experiment. The use of the steel ruler for the presence of the length of the sample is not the best way. It would be better to provide full geometry information for all the samples, so if anyone would like to reproduce the experiment it would be possible. You can always provide the full number of the standard used for the samples. There is no ISO-A80 or ISO-A50 standards. Did you mean the 6892-1 standard used for the tensile test in the room temperature?
2/ How were the samples for the DIC analysis prepared? Did some points were used on the surface for the software to fallow, or the software has digital points included? There is lack of such information in the manuscript.
3/ In the manuscript the authors write: The reasons for this distribution was discussed by Tasan et.al. [22]: that sharp deformation bands nucleate at ferrite grains and propagate in the softest route within the microstructure with angles of 45–50 to loading direction. Do the authors have the proof that in case of DP800, the mechanism is the same? For example, some microstructure investigation?
4/ Fig. 6. Which sample is presented on the picture? A80, A50, mini…?
5/ Fig. 7 and 9 it would be better to include the legend in the figure. It would be easier to see, which curve is for which sample.
6/ Fig. 12. Why only two result have error bars? There are 5 experimental values on the picture and three of them are without the error? What is the reason for such presentation of results?
7/ Why the amount of A50 samples in much grater then for the A80 (fig. 13)? What about the mini samples, how many samples were tested? There is no information about this in the manuscript. Is it even comparable if the amount of samples differ so much?
8/ Please select on a way to describe the pictures. Use a); b) c)…. Or left/right. One time the left/right description is used, another the a); b)….
Round 2
Reviewer 2 Report
The article has improved after the revision process.
Reviewer 3 Report
The authors have fully-addressed the comments furnished by the Reviewer satisfactorily. Therefore, this paper can now be accepted for publication in this top journal.
Reviewer 4 Report
Thank you for the changes in the manuscript according to the reviewers suggestion. Right now, in my opinion the manuscript is sufficient for the publication.